

# Antigenic properties of the SARS-CoV-2 nucleoprotein are altered by the RNA admixture

Denis E. Kolesov[1], Maria V. Sinegubova[1], Irina V. Safenkova[2], Ivan I. Vorobiev[1] and Nadezhda A. Orlova[1]

[1] Laboratory of Mammalian Cell Bioengineering, Institute of Bioengineering, Research Center of Biotechnology of the Russian Academy of Sciences, Moscow, Russia
[2] Laboratory of Immunobiochemistry, Institute of Biochemistry, Research Center of Biotechnology of the Russian Academy of Sciences, Moscow, Russia, Moscow, Russia

## ABSTRACT

Determining the presence of antibodies to the SARS-CoV-2 antigens is the best way to identify infected people, regardless of the development of symptoms of COVID-19. The nucleoprotein (NP) of the SARS-CoV-2 is an immunodominant antigen of the virus; anti-NP antibodies are detected in persons previously infected with the virus with the highest titers. Many test systems for detecting antibodies to SARS-CoV-2 contain NP or its fragments as antigen. The sensitivity and specificity of such test systems differ significantly, which can be explained by variations in the antigenic properties of NP caused by differences in the methods of its cultivation, isolation and purification. We investigated this effect for the *Escherichia coli*-derived SARS-CoV-2 NP, obtained from the cytoplasm in the soluble form. We hypothesized that co-purified nucleic acids that form a strong complex with NP might negatively affect NP's antigenic properties. Therefore, we have established the NP purification method, which completely eliminates the RNA in the NP preparation. Two stages of RNA removal were used: treatment of the crude lysate of *E. coli* with RNase A and subsequent selective RNA elution with 2 M NaCl solution. The resulting NP without RNA has a significantly better signal-to-noise ratio when used as an ELISA antigen and tested with a control panel of serum samples with antibodies to SARS-CoV-2; therefore, it is preferable for in vitro diagnostic use. The same increase of the signal-to-noise ratio was detected for the free N-terminal domain of the NP. Complete removal of RNA complexed with NP during purification will significantly improve its antigenic properties, and the absence of RNA in NP preparations should be controlled during the production of this antigen.

## INTRODUCTION

The coronavirus pandemic affects the life of the entire population of the globe, with 258 830 438 confirmed cases of COVID-19 for 26.11.2021. The emergence of new SARS-CoV-2 variants (*Khan et al., 2021*) escaping the immune defense and variants with increased contagiousness and the impossibility of global vaccination of the entire planet population suggest that the pandemic will continue for several more epidemic seasons.

Corresponding author
Nadezhda A. Orlova,
nobiol@gmail.com

There is still a need for mass and cost-effective production of tests. This problem is especially acute in countries with underdeveloped economies. The widespread availability of serological tests will help defeat the pandemic and return to normalcy.

Acute or past infection of the individual with the SARS-CoV-2 virus may be tested using various *in vitro* diagnostic methods, mainly the RT-PCR or serological testing for viral antigens presence or anti-viral antibodies (*Kilic, Weissleder & Lee, 2020*). Testing for anti-SARS-CoV-2 antibodies reliably indicates the presence of anti-viral immunity, regardless of the presence or absence of symptoms of the disease. The generalized data from the population-wide serological studies makes it possible to accurately assess the transmission dynamics and the actual scale of the virus's spread (*Dorigatti et al., 2021*). During mass vaccination and for quite a long time after it, serological tests will also play a critical role in assessing the effectiveness and duration of the vaccine's action.

Potential antigens for SARS-CoV-2 serological tests were evaluated on their specificity against other common human coronaviruses (HCoV-OC43, HCoV-HKU1, HCoV-229E, HcoV-NL63) (*Tilocca et al., 2020*; *Dobaño et al., 2020*) and it was found that among the four viral structural proteins (proteins S, E, M, N), two most abundant proteins –S and N (NP) are sufficiently specific; they are most often used for practical serological testing. The N-antigen is the most abundant proteinaceous component of the viral particle; therefore, it can be assumed that N-antigen causes the appearance of the highest antibody titers during the infection. Serological tests using the N antigen are expected to have the best sensitivity. According to *Okba et al. (2020)*, ELISA with the S1 antigen was more specific in discriminating COVID-19 patients and HCoV or MERS-CoV groups, and ELISA with the NP was more sensitive in revealing the seropositivity of patients with mild COVID-19. In direct comparison, the S antigen gives slightly higher ELISA values than the NP antigen for the COVID-19 patients (*Chen et al., 2020*).

Most of the approved and used to date SARS-CoV-2 vaccines do not elicit a humoral immune response to NP (*Martínez-Flores et al., 2021*); therefore, serological tests based on both S and N antigens may help to distinguish between vaccinated and/or people infected with SARS-CoV-2. The specificity and sensitivity of NP-based serological tests should be as high as possible since S-antigen tests cannot be used as a parallel or control method for detecting SARS-CoV-2 infection in a significant proportion of patients.

The nucleoprotein of beta coronaviruses (NP) is immunodominant and most abundantly expressed viral protein in the cytoplasm of infected cells (*Timani et al., 2004*). It consists of two tightly folded domains, N-terminal (NTD) and C-terminal (CTD or N2b), responsible for RNA binding and NP dimerization, respectively, and the surrounding short unstructured regions N1a, N2a, and B/N3. Nucleoprotein is the only cytoplasmic protein of coronaviruses that is purposefully captured by viral particles separating from infected cells. The NP is synthesized in the cell cytoplasm, so it is not glycosylated. It contains a single cluster of Ser/Thr phosphorylation sites in the N2a subdomain (*Tugaeva et al., 2021*), and, probably, some NP molecules have acetylated Lys residues (*Hatakeyama et al., 2021*), so the recombinant NP from *Escherichia coli* will adequately represent the natural NP as the antigen. At the same time, the recombinant NP, produced in mammalian or yeast cells as the secreted protein, contains multiple N- and O-glycan moieties and,

presumably, is partially shielded from binding antibodies (*Supekar et al., 2020*). *E. coli* derived NP, its single domains, and deletion mutants are widely used in serological assays (*Guo et al., 2020*; *Yamaoka et al., 2021*).

NP is a highly basic protein prone to multimerization and non-specific binding of nucleic acids (*Zeng et al., 2020*), and both the NTD and CTD contain highly positively charged regions (*Chang et al., 2014*). Residual admixtures of host cell biopolymers and polymerization state of the NP could significantly change depending on its purification from specific impurities; therefore, it is necessary to develop a method for obtaining a SARS-CoV-2 NP preparation that provides the maximum efficiency of the serological testing. Many publications (*Nasrallah et al., 2021*; *Tehrani et al., 2020*) mention the highly differing quality of commercially available NP-based tests. Still, they do not discuss specific impurities in the antigen that can cause a decrease in the sensitivity of the tests or methods for specific purification of these impurities (*Liu et al., 2021*).

We proposed that NP's antigenic properties may be significantly altered due to the presence of nucleic acid impurities, which will force the NP to the multimeric state (*Perdikari et al., 2020*), partially block its surface, and expel some of the NP-binding antibodies by the negatively charged patches. Methods of nucleic acid removal from viral nucleoproteins are well known—treatment with nucleases and dissociation of complexes with the high salt solutions (*Damodaran & Kinsella, 1983*). Here, we present the simple protocol of SARS-CoV-2 NP expression and purification and demonstrate the change in its antigenic properties upon two-stage removal of nucleic acid contaminants.

## MATERIALS & METHODS

### Molecular cloning

The NP open reading frame flanked with the *Pci* I-*Hind* III sites was synthesized by Epoch Life Science, Inc (Missouri City, TX, USA). The synthetic gene was cloned into the pHYP expression vector (Genbank ID: MW187859), cut at the *Nco* I and *Hind* III sites. The codon usage of the ORF was optimized for *E. coli* expression with the GENEius-Light software (https://geneius.de/GENEiu), and a C-terminal 10xHis cluster was added to the ORF for further purification of the target protein by metal chelate chromatography. The resulting expression plasmid, pHYP-NPC-10H (Genbank ID: MW187860, Addgene #162789), was transformed into the *E. coli* BL21[DE3] cells.

The *Avr*II and *Nhe*I sites were introduced into the synthetic NP ORF using the Silent Mutator online tool (https://www.molbiotools.com/silentmutator.html).

The N-terminal part of NP ORF was deleted using the *Avr*II and *Nhe*I restriction of the pHYP-NPC-10H plasmid with subsequent ligation, resulting in the pHYP-NP-tail-C-10H plasmid, the protein coded corresponds to residues 218-419 of the SARS-CoV-2 NP, protein code—CTD.

The C-terminal part was deleted using *Nhe*I-*Hind*III restriction of the pHyp-NP plasmid and exchanged to the pair of annealed oligos AD-NPhead-NheF (5′-CTAGCCCATCACCATCATCACCACCATCACCATCACTGATA-3′); AD-NPhead-HindR (5′-AGCTTATCAGTGATGGTGATGGTGGTGATGATGGTGATGGG-3′),

resulting in pHYP-NP-head-C-10H plasmid, the protein coded corresponds to residues 1–220 of the SARS-CoV-2 NP, protein code –NTD. Sequences of the expected proteins are shown in Fig. S1.

## Bacterial culture

Expression plasmids were transformed to the BL21[DE3] cells. The resulting strains were grown in shake flasks in the 2xYT medium with 30 mg/L kanamycin and 0.1% glucose at 37 °C, induced with the 1 mM isopropyl-$\beta$-D-thiogalactopyranoside at OD600 0.6-1 AU and cultivated at 30 °C for 3 h.

Bacterial cells were collected by centrifugation, 1 g of wet cell paste obtained was resuspended in 10 ml lysis buffer (20 mM sodium phosphate, pH 7.4; 10 mM imidazole, 10 $\mu$g/ml egg white lysozyme, 0.1% Triton X-100) and incubated 15 min on ice. Bacterial DNA was sheared with two sonication rounds of 20 sec using a Bandelin Sonopuls ultrasonic homogenizer (BANDELIN electronic GmbH & Co. KG, Germany). The bacterial lysates were cleared by centrifugation at 45,000$\times$ g for 10 min and then divided into two parts, one of which was treated with 10 $\mu$g/ml RNAse A (TThermo Fisher Scientific, Waltham, MA USA) for 15 min on ice, and the second one was left untreated (NP-RNA; NTD-RNA).

## NP and NTD purification

Immobilized metal affinity chromatography was carried out at room temperature on the Akta Explorer system and the Tricorn 5/5 column, packed with 1 ml of Ni-Sepharose resin (all Cytiva, Marlborough, MA USA). The optical density of the eluate was monitored and recorded using the instrument control software Unicorn (Cytiva). The column was equilibrated with 20 mM sodium phosphate, pH 7.5; 500 mM NaCl, 10 mM imidazole solution. Clarified bacterial lysates were applied to the column at 0.5 ml/min. The column was washed with equilibration solution until a stable baseline at 1 ml/min flow (approximately 10 column volumes), then with the equilibration solution with 50 mM imidazole at the same flow velocity for 10 column volumes. The column was optionally washed with the equilibration solution with 2 M NaCl (NP, NP-NaCl, NTD, NTD-NaCl samples) until the stable baseline. Target proteins were eluted with the equilibration solution with 250 mM imidazole-HCl at 0.5 ml/min; main eluate peaks were collected as single fractions. Ten minutes flow stop was performed during elution for the yield maximization. Purified proteins were divided into aliquots, flash-frozen in liquid nitrogen, and stored frozen for further use. Purified protein solutions were intentionally not desalted from NaCl and imidazole and used in further experiments as is due to rapid precipitation of the NP upon desalting to the PBS solution. Target protein concentration was determined using the Bradford method with the Total protein kit (Sigma, St. Louis, MO, USA).

## Nucleic acids concentration measurement

The concentration of RNA and DNA in protein samples was measured using the Qubit fluorometer (TThermo Fisher Scientific) and Qubit RNA and Qubit DNA HS reagent kits.

## SDS-PAGE

Proteins were analyzed by SDS–PAGE (12.5% acrylamide) in reducing conditions with the PageRuler prestained marker (Thermo Fisher Scientific), 5 $\mu$l/lane. The gels were

stained with colloidal Coomassie blue, de-stained with the deionized water until clear background, scanned using the flatbed scanner in the transparent mode, saved as 16-bit grayscale images. The images were analyzed using the TotalLab TL120 software (Nonlinear Dynamics, Newcastle upon Tyne, UK).

## Analytical size-exclusion chromatography

Size exclusion chromatography was performed according to *Sinegubova et al. (2021)* with minor changes. Mobile phase was 20 mM sodium phosphate pH 7.5, 100 mM imidazole-HCl, 150 mM or 300 mM or 2 M NaCl. Size exclusion chromatography with the multi-angle light scattering (MALS) detection was performed as described in *Tugaeva et al. (2021)* with minor changes –mobile phase was 20 mM sodium phosphate pH 7.5, 100 mM imidazole-HCl, 300 mM NaCl, flow 0.5 ml/min, column Superdex 200 10/300 (Cytiva).

## Dynamic light scattering (DLS)

The hydrodynamic diameter (Dh) of proteins was measured using DLS as described in (*Safenkova et al., 2016*) with minor changes. Protein concentrations were in the 0.6–1 mg/ml range for all samples tested, 20 mM sodium phosphate pH 7.5, 500 mM NaCl, 250 mM imidazole solution in all cases.

## Enzyme-linked immunosorbent assay (ELISA)

Control pooled serum samples with established immunoreactivity in clinically approved tests were obtained from Anti-SARS-CoV- 2 Verification Panel for Serology Assays (NIBSC panel, National Institute for Biological Standards and Control, Potters Bar Hertfordshire EN6 3QG, UK). The NIBSC Panel is comprised of 37 samples, 23 positive samples, and 14 negative samples. We also used the samples described in the article (*Sinegubova et al., 2021*), namely pre-COVID-19 normal human plasma sample (Renam, Moscow, Russia) and control pooled serum samples obtained from patients with the PCR-confirmed SARS-CoV-2 infection (Xema Co., Ltd, Moscow, Russia). All samples were tested in triplicates if not stated otherwise.

Microtiter plate wells (Corning, NY, USA) were coated with 100 µl of 1 µg/ml test antigen in PBS with 100 mM imidazole solution and incubated overnight at + 4 °C. The wells were washed three times with PBS–0.02% Tween (PBST) and blocked for 1 h at 37 °C with 3% bovine serum albumin (BSA; Sigma) in PBS, washed with PBST, and used immediately. Test sera were prediluted with 1% BSA-PBS, applied as serial twofold dilutions in the 1:400–1:25 600 range, and incubated for 1 h at 37 °C. Wells were washed three times with PBST, secondary anti-human IgG antibody-HRP conjugate (Xema Co., cat. T271X@1702) was used at the 1:20 000 dilution. After 1 h at +37 °C, wells were washed five times with the PBST and 100 µl of ready-to-use TMB solution (Xema Co.) was added to each well. The color was developed for exactly 10 min at room temperature (+25 ± 2 °C). The reaction was stopped by addition of 100 µl of 5% orthophosphoric acid per well. Absorbance at 450 nm was measured with a Multiskan EX plate reader (Thermo Fischer Scientific).

## Mass spectrometry

High-performance liquid chromatography-electrospray ionization-mass spectrometry (HPLC-ESI-MS) analysis was performed with the Impact II QqTOF high-resolution mass-spectrometer (Bruker Daltonik, Germany), the UHPLC (Bruker Daltonik), and the Nucleodur C4 300-5 ec 4.6*150 mm column (Macherey-Nagel, Germany). Data acquisition conditions were as follows: flow 1 ml/min, split 1:10, gradient elution from 2% to 98% B in 40 min (A: 0.1% formic acid in water, B: 0.1% formic acid in acetonitrile), column temperature 40 °C, 30–50 µg of sample per injection, ESI source in positive mode, HV capillary at 4.5 kV, spray gas—nitrogen at 1.8 bar, dry gas—nitrogen at 8 L/min 220 °C. Spectra were processed with BioPharma Compass 3.1.1 (Bruker Daltonik, Germany).

## Statistical analysis

ELISA data were analyzed by the Student's two-tailed unpaired $t$-test or one-way ANOVA with the post-hoc Tukey-Kramer HSD test utilizing the GraphPad Prism (GraphPad Software, San Diego, CA, USA), LibreOffice (https://www.libreoffice.org/) and the RStudio software (RStudio PBC, Boston, MA, USA). All Student's and ANOVA $p$-values are presented with the raw OD readings data in the Supplemental Information.

Correlational analysis for ELISA data obtained for the NIBSC panel serum samples was performed for five semi-quantitative and quantitative assays (published results) and two experimentally performed assays with full-length NP antigen variants. The analysis was performed for all 37 serum samples and the COVID-19 positive sera subset. Pearson's r was calculated using the RStudio software. In the case of Pearson's r 0–0.19 is regarded as very weak, 0.2–0.39 as weak, 0.40–0.59 as moderate, 0.6–0.79 as strong, and 0.8–1 as very strong correlation. Scripts employed for the correlation analysis are deposited at the GitHub website: https://github.com/d-kolesiko/ELISA.

# RESULTS

## Expression and purification of the intact NP

*E. coli* strain BL21 [DE3] was transformed with the plasmid pHYP-NPC-10H encoding the full-length NP SARS-CoV-2 with a C-terminal 10xHis tag (Fig. 1A). During induction at 30 °C, abundant expression of the target protein in a predominantly soluble form was observed (Fig. 1B); therefore, further optimization of the cultivation conditions was not carried out. The target protein was purified from the supernatant of lyzed bacterial cells by the immobilized metal affinity chromatography (Fig. 1C). According to UV absorption data, the purified NP contained a significant amount of nucleic acid admixture - 280:260 nm absorption ratio was 0.514, far below the expected 1.75 ratio of the pure protein. According to the intercalating dye fluorescence intensity analysis performed with the Qubit fluorimeter (Thermo Fischer Scientific), contaminating nucleic acids were found to be the RNA. RNA:protein mass ratio was 25.0 % for the NP preparation, untreated with the RNAse A (NP-RNA protein sample). Lysis and purification of the full-length NP were repeated with two additional RNA removal steps—treatment of the crude cell lysate with the 10 mg/L RNAse A for 15 min on ice and column wash with the equilibration solution, adjusted to the 2 M NaCl concentration. According to the chromatography traces data

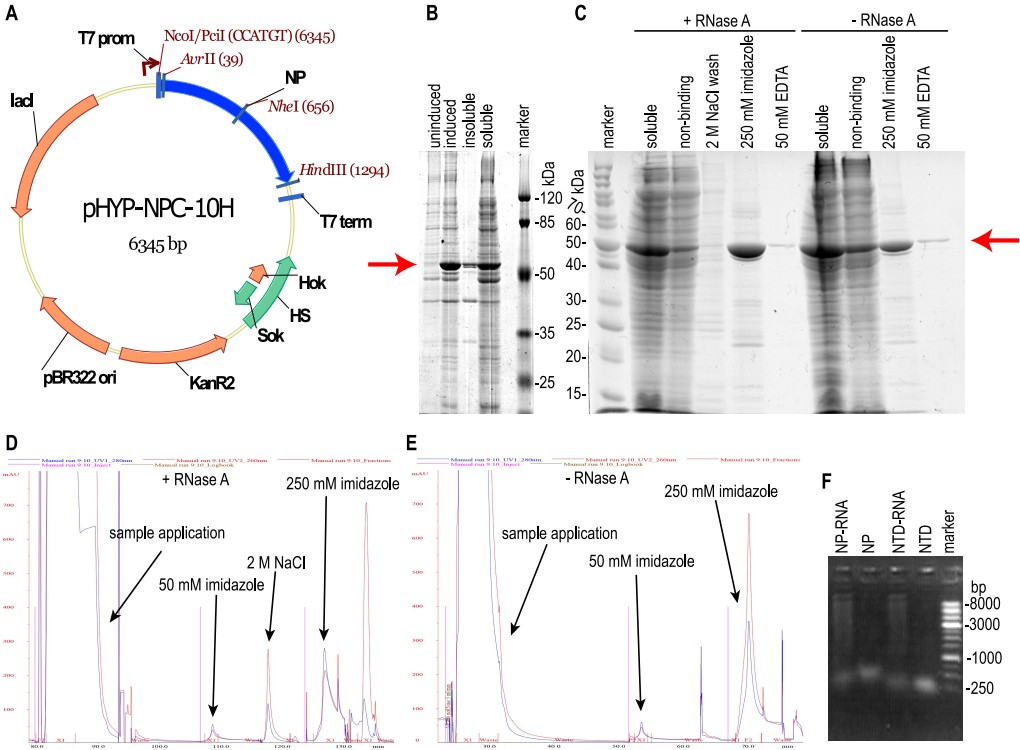

**Figure 1   Expression and purification of the full-length SARS-CoV-2 nucleoprotein.** (A) Map of the pHYP-NPC-10H plasmid. 10xHIS—C-terminal decahistidine tag; KanR2—kanamycin A resistance gene; HS—hok/sok post—segregation killing locus; lac I—gene of the LacI repressor protein; T7 prom—promoter for bacteriophage T7 RNA polymerase. (B) SDS-PAGE analysis of the full—length NP expression. Induction with 1 mM IPTG. (C) SDS-PAGE analysis of the NP purification in the presence or absence of the RNase A. Target protein position is marked with the red arrow. "Non-binding"—column flowthrough fraction. (D, E) IMAC chromatography traces for the NP purification in the presence or absence of RNase A treatment and 2 M NaCl solution wash. Eluate absorbance at 280 nm is in blue, absorbance at 260 nm—in red. (F) Agarose gel analysis of purified proteins, ethidium bromide staining, double-strand DNA marker.

(Figs. 1D, 1E), 2 M NaCl solution was sufficient to dissociate the protein— RNA complexes and elute the contaminating RNA. Subsequent elution of the target protein with the 250 mM imidazole solution produced the elution peak with the 280/260 nm absorption ratio of 1.2, typical for the pure protein. Imidazole elution of the target protein without the RNAse A treatment and 2 M NaCl wash produced a peak with the 280/260 nm absorption ratio of 0.53, indicating the significant contamination with nucleic acids. This observation is in line with the intercalating dye analysis of NP preparations. The stepwise gradient of the NaCl concentration in the wash solution revealed that admixtures with the 280:260 nm light adsorption ratio 1:2, typical for the nucleic acid, are eluted mostly with the 1 M and 1.5 M NaCl solutions, making the 2 M NaCl solution sufficient for the complete removal of nucleic acids from the RNase A – treated NP (Fig. S2).

It should be noted that the NP preparation, purified from RNA with both RNAse A treatment and high salt column wash, was completely RNA-free; residual RNA level was below the detection limit of the Quant-iT$^{TM}$ RNA Assay Kit used, *i.e.*, below the 0.05% (w/w). According to this measurement, the NP was purified from the RNA at least 12-fold. Both samples of purified NP were tested for the DNA admixture using the Quant-It DNA high sensitivity dye and were found to be DNA-free, residual DNA level was below 12 ppm in both samples. We made two additional control NP preparations—one treated with the RNAse A and not washed on-column with the 2 M NaCl, another—washed with 2 M NaCl and untreated with the RNAse A, and used these preparations for ELISA. Nucleic acids in the NP preparations visualized on the agarose gel (Fig. 1F) turned out to be highly polymeric.

### Expression and purification of the NTD and CTD fragments

NTD and CTD of the NP in the pHYP plasmid were transformed into the *E.coli* BL21[DE3] cells. Producers were cultured in the same way as for the full-length NP. We did not detect the protein band corresponding to the CTD upon induction even in the insoluble proteins fraction (Fig. S3). At the same time, free NTD was overexpressed similarly to the full-length NP (Fig. 2A) and was purified with or without the RNA removal steps as was described above for the full-length NP (Fig. 2B).

### NP oligomerization state analysis

According to the ESI-MS data, the molecular mass of the purified full-length NP, treated with both RNase A and 2 M NaCl wash, was 46866.94 Da, in good agreement with the expected molecular mass of the NP without the N-terminal Met residue; expected average molecular mass 46865.95 Da (Figs. 2C, S4). Similar data were obtained for the NTD protein, treated with both RNase A and 2 M NaCl wash (Figs. 2D, S5), molecular mass obtained 24637.52 Da, expected molecular mass 24637.81 Da for the NTD without N-terminal Met. The relative mobility of the NP on the SDS-PAGE corresponds to the 52.7 kDa, indicating the slight gel retardation typical for the His-tagged protein.

The oligomerization state of the full-length NP preparations was checked by the size-exclusion chromatography in physiological (Fig. S6) and high-salt solutions (Figs. 2E, 2F, S6). We observed the anomalous retention time of the target protein on the Superdex-200 column in physiological conditions in the presence and in the absence of contaminating RNA (Fig. S6). In the presence of 300 mM NaCl in the mobile phase, the RNA-free NP was present mainly in the dimeric form according to the data of MALS detector - 82.4 kDa (Fig. S7A), but the retention time of the main peak corresponded to the 142 kDa with the minor monomer peak (43.6 kDa) visible (Fig. 2E). This anomalous behavior of the SARS-CoV-2 NP was already described in other studies (*Tugaeva et al., 2021*) and may correspond to the complex shape of the NP dimer. At the same time, the NP-RNA produced multiple minor peaks corresponding to various oligomerization numbers, and the major peak was heavily retarded on the column and was not suitable for molecular mass calculation using the direct MALS detection or retention time curve interpolation (Fig. 2F). Further increase in the ionic strength of the mobile phase to 2 M NaCl resulted in

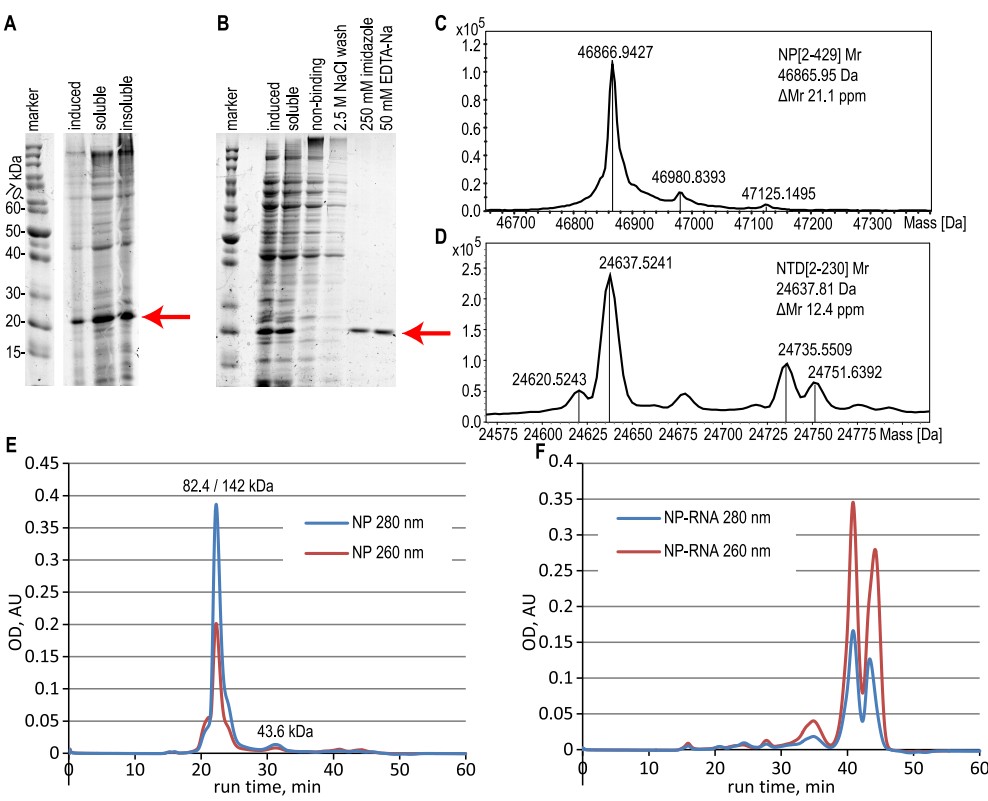

**Figure 2** **Expression and purification of the N-terminal domain of the nucleoprotein, analysis of the full-length NP and NTD.** (A) SDS-PAGE analysis of the NTD expression, induction with 1 mM IPTG. (B) SDS-PAGE analysis of the NTD during the purification with the RNase A treatment. Target protein position is marked with the red arrow. (C, D) ESI-MS mass spectra of the full-length NP and NTD. (E, F) Size-exclusion chromatography traces of the pure NP and NP-RNA, absorbance detection at 280 nm (red) and 260 nm (blue). Mobile phase–500 mM NaCl, 20 mM sodium phosphate, 100 mM imidazole. The protein molecular mass in the main peak was determined by the MALS detector (left value) and calculated from the calibration curve (right value), the molecular mass of the right peak was calculated from the calibration curve.

the retardation of the major peaks for both NP samples due to the hydrophobic interaction of the NP and the chromatography sorbent (Figs. S6E, S6F). NTD protein, unlike the full-length NP, was present mostly in the monomeric form, according to both MALS data and calibration curve interpolation (Figs. S7B, S10) in the physiological conditions and in the presence of 300 mM NaCl (Figs. S10A–S10D). Similar, but not the same, resin retardation results were obtained for the NTD fragment– in the presence of 2 M NaCl the RNA-free NTD was seen as multiple retarded peaks, including the normal monomeric form, and in the presence of RNA only the retarded forms and small quantities of non-retarded highly polymeric form of the NTD were seen; monomeric NTD is completely absent on the chromatography trace (Figs. S10E, S10F).

The SEC analysis of NP and NTD variants shows that both proteins form various multimeric complexes, apparently interacting with the chromatography resin at all NaCl

concentrations used, so all protein samples, including the partially purified NP and NTD variants, were subjected to the dynamic light scattering analysis in solution.

Both completely purified NP and NTD proteins contained a mixture of small and large protein particles. Nucleoprotein molecules were distributed nearly equally between 18 nm particles (radius corresponds to the dimeric form) and large particles with $D_h > 1000$ nm (Figs. S8A, S8B), and most of the NTD molecules are present in the presumably monomeric form, $D_h$ 9.2 nm (Figs. S9A, S9B). Ribonuclease A treatment without the on-column 2 M NaCl wash was sufficient to remove the intermediate particles in the case of NP protein; particle median sizes and mass distribution are very close for NP and NP RNase+, NaCl- preparations (Figs. S8C, S8D). At the same time, the NTD monomer was not seen after RNase A treatment, most NTD molecules remained in the highly aggregated form. Sodium chloride wash without the RNase A treatment was also insufficient for generating small protein particles in the concentrated NP or NTD solutions (Figs. S8E, S8F, S9E, S9F).

For the NP-RNA preparation that had not undergone RNase or 2 M NaCl treatments, only large particles were observed, forming the dominant population of particles with $D_h > 1000$ nm (Figs. S8G, S8H) and a high % polydispersity (%Pd), equal to 70.4%. Oligomeric particles are not seen in this population at all. The NTD-RNA protein sample also was found to contain mainly highly polymeric protein particles, with $D_h > 1000$ nm and %Pd 8.4, but there was a peak with an average particle size of 77 nm (%Pd 5.7%) (Figs. S9G, S9H). These results confirmed that only a combination of RNase A treatment and on-column 2M NaCl wash is sufficient for depolymerization of the NP, which should maximize the surface accessibility for binding with antibodies.

Oligomerization of the *E. coli*-derived NP from the closely related beta-coronaviruses SARS-CoV and MERS-CoV has been previously investigated in several works: (*He et al., 2004*; *Cong et al., 2017*), and it was found that NP tends to change the oligomerization state in the presence of nucleic acids. Salt effects on the SARS-CoV-2 NP polymerization were not studied before, but for the unrelated DNA-binding protein of the herpesvirus VP19C, the formation of the protein-DNA complex and the subsequent polymerization are inhibited by increase of the NaCl concentration to 1 M (*Bera et al., 2014*). Although a complete understanding of the mechanism of interaction of nucleoprotein monomers with each other has not been achieved, most of the data indicate that NP exists in the form of a multimer in the presence of nucleic acid and in the dimeric form after extensive purification (*Perdikari et al., 2020*).

## ELISA

Full-length NP preparations and NTD preparations were used as the immobilized antigen in the antibody capture ELISA testing of the anti-SARS-CoV-2 antibody titers. Proteins were applied to the microplates in exactly equal amounts; the protein concentration was determined in all cases using the Bradford method, which is insensitive to the RNA admixture. Dose–response curves were obtained for positive and negative pooled serum samples, and it can be clearly seen that RNA-free NP antigen gives a significantly (Student's one-tailed paired *t*-test, all ELISA data analyzed using this test, if not stated otherwise) higher signal at various dilutions of the COVID-19-positive pooled sera (Fig. 3A).

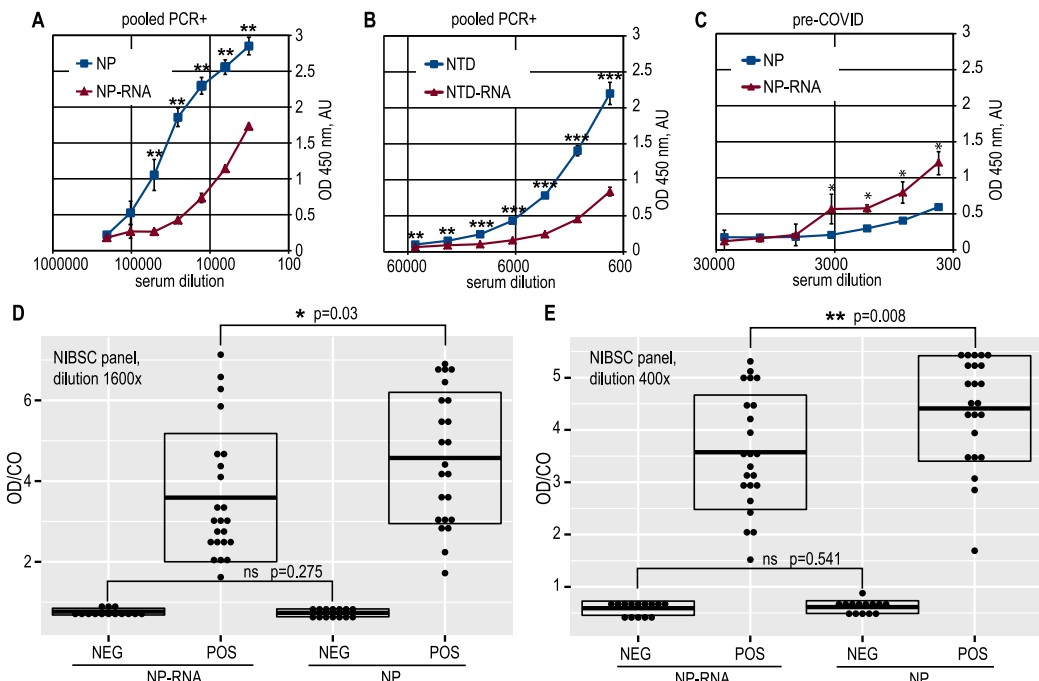

**Figure 3 Antigenic properties of the NP and NTD proteins.** (A) Antibody capture ELISA with the pure NP and NP-RNA antigens, pooled PCR+ sera. (B) Antibody capture ELISA with the pure NTD and NTD-RNA antigens, pooled PCR+ sera. (C) antibody capture ELISA with the pure NP and NP-RNA antigens, pooled pre-COVID sera. (D) Positivity indices of the control serum samples from the NIBSC sera panel, samples dilution 1600x; NEG, negative samples; POS, positive samples. (E) Same to (D), samples dilution 400x. Statistical analysis by the paired $t$-test for all panels, $n = 3$, * - $p < 0.05$; ** - $p < 0.01$; *** - $p < 0.001$.

A similar effect was retained for the NTD antigen (Fig. 3B). At the same time, the pooled pre-COVID-19 sera, derived in-house from the lyophilized normal blood plasma, was significantly more reactive with the RNA-complexed full-length NP (Fig. 3C) than with the pure NP protein. The non-specific binding of the antibodies in the pre-COVID pooled sera to the NTD antigen was very similar for pure NTD and NTD-RNA. However, the NTD gave a much weaker ELISA signal than the full-length NP for the COVID-19-positive pooled sera, resulting in a lower overall signal-to-noise ratio, determined as 10,9 for the NP and 8,9 for the NTD (Figs. S11F, S12E). Further analysis was carried out for the full-length NP as the more sensitive and specific antigen.

We also tested all partially purified NP and NTD preparations –treated with the RNase A alone or with 2 M NaCl column wash alone (Figs. S11, S12). Expectedly, partial antigen purification resulted in diminished assay signal-to-noise ratios in both cases, mainly due to lower antibody binding to the RNA-complexed antigens for the pooled PCR+ sera.

ELISA testing of the pure NP and NP-RNA antigen was performed for all serum samples from the NIBSC control COVID-19 sera panel, tested at two dilutions –1:400 and 1:1600 (Figs. 3D, 3E) ($p = 0.008$ and $p = 0.03$ respectively, Welch Two Sample $t$-test, RStudio). Positivity indices were calculated as the ratio of the optical density for the test sample and the mean optical density in negative samples wells plus three times the standard

deviation, according to the *Lardeux, Torrico & Aliaga (2016)* (OD/CO ratio). For both dilutions tested, the statistically significant difference of the OD/CO ratio was detected for the positive samples group—RNA-free NP antigen was significantly more reactive with antibodies from positive samples. At the same time, the OD/CO ratio for negative samples was very similar for both sera dilutions tested.

We tested the correlation of the OD/CO ratios obtained for NIBSC control panel, pure NP antigen, NP-RNA antigen, and published signal strengths for several clinically approved serological tests (Supporting tab1). All datasets were strongly correlated with the Pearson's $r > 0.8$ (Fig. 4B) (*P*-values are approximated using the t or F distributions). Correlation analysis was performed again for the positive subset of the NIBSC control panel, and it was found that both pure NP and NP-RNA antigens still produce highly correlated results, Pearson's $r > 0.8$. At the same time, correlation of OD/CO for our test results with the quantitative and semiquantitative results of few clinically approved test systems was lower, $0.4 < r < 0.8$, and correlation of experimentally obtained ELISA data and published data for S antigen and its fragments was approximately the same to the correlation of experimentally obtained data and NP-based approved tests data (Fig. 4A) (*P*-values are approximated using the t or F distributions). Generally, the correlation of quantitative results of various tests for positive samples from the NIBSC control panel gives the Pearson's r in the 0.4–0.8 range. Similar Pearson's r values were found for three various automated serological tests in the case of real-world datasets –serum samples from the Qatar population (*Nasrallah et al., 2021*).

Thus, the RNA-free SARS-CoV-2 NP obtained as described in this article is expected to be more suitable for serological assays than the conventionally purified NP containing the tightly bound RNA. Two different techniques of RNA removal should be performed; neither alone is sufficient for the complete purification of the NP antigen.

## DISCUSSION

It is generally believed that NP and S-protein of the SARS-CoV-2 are the most specific antigens for serological testing. In some studies, the NP is claimed as the more specific antigen (*Turbett et al., 2021*), other studies challenge the specificity of the NP-based automated assays by the cross-antigen testing of the samples with borderline anti-NP signals (*Rosadas et al., 2020*) or by the direct comparative testing of the NP-based and S-protein RBD-based assays (*Zonneveld et al., 2021*). Testing large amounts of serum samples with both NP and S antigens allows decreasing the proportion of false determinations to less than 1% (*Poljak et al., 2021*).

A significant proportion of the approved anti-SARS-CoV-2 vaccines codes or contains only the S-protein, so the vaccinated individuals are expected to be S-protein positive regardless of the COVID-19 disease. If infected with the SARS-CoV-2, such individuals may still be reliably screened using the NP-based serological tests. Although the general method of obtaining the recombinant beta-coronavirus NP is well known (*Timani et al., 2004*), and the presence of the RNA admixture in the NP was also described previously (*Liu et al., 2021*), alteration of its antigenic properties caused by the RNA impurity was not

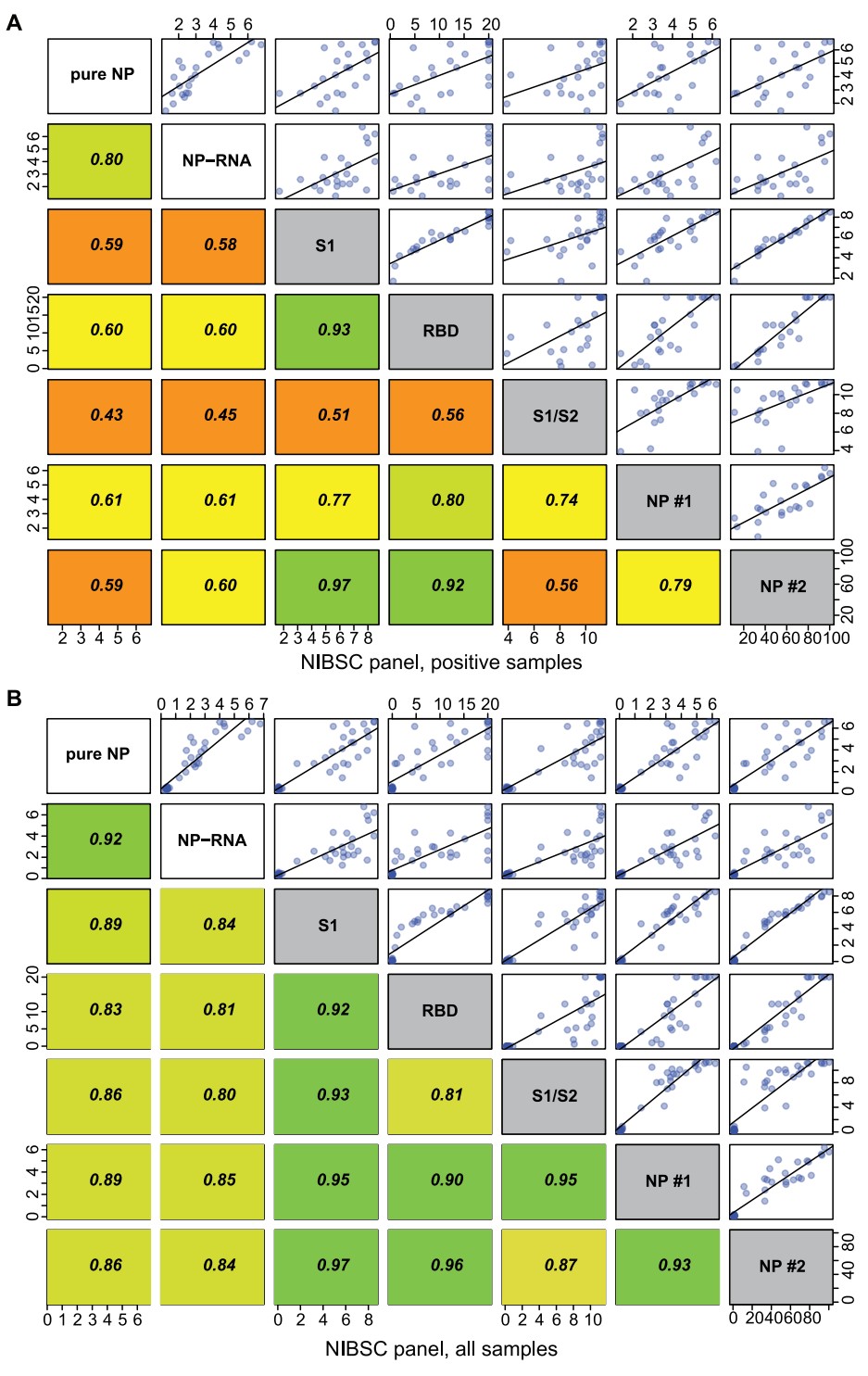

**Figure 4  Correlation analysis of the NP ELISA and reference assays for NIBSC control sera samples.** The plot shows the Pearson coefficient that was used for the estimation, as well as the linear approxima-tion. Pure NP and NP-RNA–experimentally performed assays, other data were taken from the NIBSC control sera panel description. S1, RBD, S1/S2 NP #1, NP #2 assay codes are described in detail in the Table S1. (A) Correlation analysis for the positive sample's subset; (B) correlation analysis for all samples.

reported yet. We have demonstrated that the NP as the ELISA antigen dramatically loses its immunoreactivity upon binding the bacteria-derived RNA.

We have shown that the antigenic properties of the SARS-CoV-2 nucleoprotein change significantly upon the complete removal of the associated RNA. In ELISA tests with the RNA-free NP antigen, we observed a statistically significant increase in the observed positivity indices for control positive sera and virtually no change in the nonspecific antigen binding for the negative control sera.

When testing a pooled serum sample obtained from PCR-confirmed patients and a pre-COVID-19 pooled serum sample, it was also found that completely RNA-free NP is significantly less immunoreactive against a known negative sample than NP with an admixture of RNA.

The change of the antigenic properties of NP also reproduced for its free N-terminal domain. Thus, the partial loss of the antigenic properties of the RNA-bound NP cannot be explained by the rearrangement of NP into multimeric complexes alone. Apparently, a significant part of the NP's surface is blocked by nucleic acids from contact with antibodies. It can also be assumed that the actual level of extracellular RNA in the studied serum samples might influence the observed titer of antibodies to NP SARS-CoV-2; a more detailed study of this issue may be the subject of further research.

## CONCLUSIONS

We have demonstrated the importance of specific purification of NP preparations for serodiagnostic testing from co-purified nucleic acids and identified one of the possible causes for the variation in the results of different serological tests. We have developed a purification procedure that includes only the linearly scalable operations and may be used without major changes to produce the desired quantities of NP, suitable for serological testing. We believe that the imperfect specificity of NP-based serological tests for SARS-CoV-2 could be associated with insufficient antigen purification from RNA. The specific removal of nucleic acids during NP production would improve the selectivity of new and existing serological tests.

## ACKNOWLEDGEMENTS

We thank Alexander Ivanov (Institute of Molecular biology Russian Academy of Sciences, Moscow, Russia) and Valentin Manuvera (Research Institute for Physico-Chemical Medicine, Moscow, Russia) for valuable comments and discussions on NP properties; Dr. Yuri Lebedin, Eugenia Kostrikina, and Xema Co., Ltd., Moscow, Russia for providing control sera samples. We thank Nikolay N. Sluchanko (A.N. Bach Institute of Biochemistry, Federal Research Center of Biotechnology of the Russian Academy of Sciences, Moscow, Russia) for the size-exclusion chromatography with MALS detection analysis and valuable comments on the NP behavior in solutions; Andrey M. Tsedilin (Institute of Biochemistry) for the ESI-MS analysis. The measurements were carried out in the Shared-Access Equipment Centre "Industrial Biotechnology" of the Research Center of Biotechnology of the Russian Academy of Sciences. DNA sequencing was carried out in the inter-institutional

Center for collective use "GENOME" IMB RAS, organized with the support of the Russian Foundation of Basic Research.

### Funding

The study was supported by the Russian Foundation for Basic Research (grant 20-58-55001) and the Ministry of Science and Higher Education of the Russian Federation. The funders had no role in study design, data collection and analysis, decision to publish, or preparation of the manuscript.

### Grant Disclosures

The following grant information was disclosed by the authors:
The Russian Foundation for Basic Research: 20-58-55001.
The Ministry of Science and Higher Education of the Russian Federation.

### Competing Interests

Ivan I. Vorobiev and Nadezhda A. Orlova are inventors of the patent RU2496877, covering the use of the pHYP plasmid.

### Author Contributions

- Denis E. Kolesov performed the experiments, analyzed the data, prepared figures and/or tables, authored or reviewed drafts of the paper, and approved the final draft.
- Maria V. Sinegubova performed the experiments, authored or reviewed drafts of the paper, and approved the final draft.
- Irina V. Safenkova conceived and designed the experiments, performed the experiments, analyzed the data, prepared figures and/or tables, and approved the final draft.
- Ivan I. Vorobiev conceived and designed the experiments, performed the experiments, analyzed the data, prepared figures and/or tables, authored or reviewed drafts of the paper, and approved the final draft.
- Nadezhda A. Orlova conceived and designed the experiments, performed the experiments, prepared figures and/or tables, authored or reviewed drafts of the paper, and approved the final draft.

### Patent Disclosures

The following patent dependencies were disclosed by the authors:
Patent RU2496877, PLASMID VECTOR pHYP WITH HIGH SEGREGATION STABILITY FOR EXPRESSION OF RECOMBINANT PROTEIN, BACTERIUM - PRODUCENT OF PRECURSOR OF RECOMBINANT PROTEIN AND METHOD TO PRODUCE RECOMBINANT PROTEIN, priority date 2011-12-15

### Data Availability

The zip-compressed bitmap images, as acquired by the scanner and camera; archived ELISA readings; archived chromatography traces are available in the Supplementary Files.
The code is available at GitHub: https://github.com/d-kolesiko/ELISA.

## Supplemental Information

Supplemental information for this article can be found online at http://dx.doi.org/10.7717/peerj.12751#supplemental-information.

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
