# Peer review of "Antigenic properties of the SARS-CoV-2 nucleoprotein are altered by the RNA admixture"

_PeerJ, doi:10.7717/peerj.12751_

## Round 0.1 · original submission · Major Revisions

Major revisions are needed as suggested by the Reviewers.

Reviewer 1 ·

Basic reporting

The manuscript by Kolesov DE et al investigates the effect of RNA admixture on antigenic properties of SARS-CoV-2 nucleoprotein (NP) in ELISA-based antibody capture assay. The authors have purified NP (and its N-terminal domain, NTD) using E. coli-based expression system and used the proteins as such or after RNA removal as antigen in the ELISA assay to detect the reactivity against SARS-CoV-2 positive serums. Authors claim that RNA-free NP is significantly better antigen in ELISA test compared to RNA-admixtured-NP.
The overall study is interesting as it studies the effect of NP’s purity on its antigenic properties. However, the manuscript in the current form suffers from several issues related to the write-up, experimental design, figures and data presentation, and interpretation of results.

1. The Introduction section of the manuscript provides an overview of current state of the art of SARS-CoV-2 tests, importance of NP as antigen for testing, and difficulties with getting pure preparations of NP. However, despite providing valuable insights, these discussions are overly long and impede with the smooth readability. The portions of the Introduction section discussing serological tests (lines 42–60), nucleoprotein (NP) (lines 61–76), NP admixtures (lines 77–89) should be re-phrased to make it concise while retaining the information provided in order to make the flow of the manuscript smooth and focused. Lines 90–94 are unnecessary since authors have not compared S antigen with N antigen in the current study. These lines could either be removed or more tightly integrated with lines 83–89 in order to make author’s motivation for the current study clearer/more sound.

2. The language used throughout the manuscript could be improved for better readability and to eliminate ambiguous/unclear sentences. For example, there are minor English problems on lines 20, 21, 26, 28, 30, 37, 41, 50, 51, 97, 99, 125, 127, 138, 140, 141, 152, 154, 155, 160, 183, 192 which can be easily fixed by using appropriate article/propositions/term.

3. Introduction section has some lines stating facts/possibilities without citing a literature reference. For example, statements in lines 34, 35 and 38 need suitable recent literature references. Similarly, statements ending or stated in lines 48, 81, 90 also need appropriate references.

4. Some of the figures are not high resolution or clear as discussed below:

4 (i). Fig 1A could be made higher resolution to avoid pixelation.

4 (ii). The resolution of Fig 1D and 1E is so poor that it is not possible to distinguish red curve from blue. Additionally, the text details on x-axis and y-axis as well as other details on top of each of the figures are in too small font to read.

4(iii). In Fig 2C, the x- and y-axes details are too small to read.

4 (iv). Fig 2E does not mention the molecular weight of the peaks.

4 (v). Fig 2D and Fig S2 are identical. It is not clear what is the purpose of authors to show the same figure twice.

4 (vi). In Fig S3 and S4, peaks have not been marked with the molecular weight.

4 (vii). Raw gel pictures in the supplementary folder have not been annotated. Therefore it is not clear which gel belongs to which experiment and what to conclude from the gels pictures.

Experimental design

1. The authors have not included any negative or positive control for their NP antigen in ELISA assay against the sera tested. The lack of a negative control (such as any globular non-NP protein) limits the specificity claim of their NP antigen. On the other hand, lack of a positive control (such as an established NP-based test antigen) makes it difficult to assess the effectiveness of their NP antigen in real-world testing scenario compared to the established NP-based tests.

2. Authors have used deca-His tag (line 100) for their NP purification unlike hexa-His tag routinely used in the protein purification community. Authors have not mentioned why they went high on His numbers. This is important because it will not only affect the net charge on NP but might also promote association with negatively charged nucleic acids. Ideally, a His-free purification system should have been employed given the high propensity of NP to associate with nucleic acids as mentioned by authors on line 83–85.

3. There are inconsistencies/missing details in the Materials and Methods section:

3 (i). The restriction/cloning sites on line 97 is PciI/HindIII, on line 99 it is NcoI/HindIII, while on Fig 1A, it is Xba1/HindIII. Authors should clarify this discrepancy in the restriction sites.

3 (ii). The amino acid sequences of NP, NTD, or CTD has not been provided. Authors should provide the complete amino acid sequences in Supplementary Information and also mention residue numbers for NTD and CTD fragments in the manuscript lines 106 and 108 respectively.

3 (iii). How were NTD and CTD purified has not been mentioned in Protein purification sub-section (lines 124–136).

3 (iv). Line 132 mentions high imidazole (250mM) for elution. Was imidazole buffer-exchanged before use as antigen?

3 (v). Gel destaining should be briefly mentioned in SDS-PAGE analysis sub-section (lines 137–142).

3 (vi). It is not clear what was the purpose of keeping high imidazole (100 mM) in size exclusion chromatography (line 145).

3 (vii). How many total positive and negative sera sample were obtained should be clearly mentioned in the ELISA subsection (lines 149–151).

3 (viii). It is not mentioned if the sera only contained IgG or had IgM along with IgG.

3 (ix). Details of UV measurements (lines 186–187) intercalating dye experiment (lines 198–199) should be provided in the Materials and Methods section.

3 (x). No mass spectrometry data has been shown for NP-NTD.

Validity of the findings

The Results section should provide additional information/elaboration of results:

1 (i). There are several lower molecular weight bands on gel in Fig 1C in 250mM imidazole lane in both gel. Clearly, author’s NP preparation has contaminants even after RNA removal. Authors should clarify how these impurities do not affect their results. The peak-width in mass spectrum in Fig 2C is pretty broad suggesting several peaks hidden in the main peak. It appears that NP preparation in the manuscript suffers from some kind of proteolysis resulting in several smaller molecular weight fragments which show up in gel and mass spectrum as pointed out above.

1 (ii). Lines 200-204, the claim that NP preparation was free from DNA and RNA has not been supported by data/fig in the manuscript.

1 (iii). Authors claim that three N-terminal amino acids are cleaved by proteases during purification (lines 214–217). Could this be avoided by use of protease inhibitors during purification?

1 (iv). Lines 234–237, authors need to discuss salt effects on NP polymerization.

1 (v). Lines 248–252 have seemingly contradictory statements.

1 (vi). In Fig S5, the ODs are not substantially different for NP and NP-RNA+NaCl curves. Lines 254–256, however mentions significant difference between the two. This suggest obvious wrong statistics done on this set of data. This should be corrected.

1 (vii). The r-values are much better for S1-NP#1, S1-NP#2, S1-RBD, RBD-NP#1 and RBD-NP#2 pairs compared to any combination of author’s NP (RNA free or otherwise) with either NP#1, NP#2 or RBD (Fig 4). This suggests that authors NP preparation discussed in this manuscript is worse than above literature reported antigens. Authors should comment on this in lines 266–279.

2. The Discussion section needs literature references for the statements in line 294 (“Although the general method….NP is well known”) and 295 (“was described previously”).

3. The claim made in line 313–314 “developed a purification procedure that is fully scalable” is not substantiated as there is no data in manuscript proving the scalability of the procedure.

In light of above concerns, the validity of findings is not beyond questions.

·

Basic reporting

In this manuscript, Kolesov et al reported that the two steps of RNA removal (RNAse A treatment of the crude E.coli lysate and subsequent selective RNA elution by the 2 M NaCl solution) results in the purification of RNA-free NP that represents better signal-to-noise ratio when used in ELISA. The author further delineates the usability of the full-length NP and N-terminal domain antigen devoid of RNA in antibody capture ELISA. They also tested the control panel of SARS-CoV-2 positive serum samples and compared the positivity of ELISA using NP or NP-RNA with other conventional antibody tests. Consequently, they found that the complete removal of RNA form NP during the purification may significantly improve its antigenic properties, and the absence of the RNA in NP preparations should be monitored during the production of antigen.
Although the article is well written and interesting, and reports what could be potentially important in COVID-19 antibody testing, the overall studies are still preliminary and proposed hypothesis is not well verified with current data. Additional experiments with proper controls are needed before the manuscript can be accepted for publication. My specific comments are listed below.

Experimental design

Figure 1C, D
Since 2M NaCl is used for RNase+, please make the same comparison for RNase-. Also, only 2M NaCl is used, please consider other concentrations. It would be possible to compare NaCl with other chaotropic salts such as NaBr, NaClO4 or Cl3COONa (https://febs.onlinelibrary.wiley.com/doi/pdf/10.1016/0014-5793(83)80675-9). Please provide the data showing the size of NP-RNA.

Figure 2A, B
This reviewer think that NP-CTD should be included as a negative control in the figure to verify the hypothesis that the authors claimed.

Figure 3A, B, C
Again, please add the data of NP-CTD as a negative control. Alternatively, authors may use other non-RNA binding proteins

Figure 4
There is no data regarding NP#1 and NP#2 tests employing NP or NP-RNA.
It would be helpful if the authors cite the reference showing the correlation between antibodies targeting NP and S1/RBD in COVID-19 patient sera.

Validity of the findings

Figure 1C, D
I'm not sure what the non-binding lane indicates. Please add an explanation.

Figure 2C
Since it is not known whether RNA exists or not in the full length NP, please add an explanation.

Figure 2D, E
In Fig1, 260nm is Red and 280nm is Blue, but the colors are reversed in this figure. Please unify the colors to make readers understand easily.

Figure 3D, E
It would be easier to see if the data of NEG and POS are described separately.

Additional comments

This article is very interesting, but I think data of NP-CTD is essential as a control.

·

Basic reporting

The manuscript needs to be improved in terms of writing. there are several grammatical errors.

current statistics about the infection and mortality can be included in the manuscript.

Experimental design

overall the study is well planned and has good results but some more experiments need to be performed to support the hypothesis.

1. Western Blot analysis to confirm the expression of nucleoproteins

2. DLS and native page analysis to further support the oligomeric state of the protein.

3. mass spectrometric analysis methods need to be included in a detailed manner along with a citation.

4. To support the ELISA result, the author can perform the dot-blot far-western analysis to find out the interactions.

Validity of the findings

this will helps in exploring the possibility of using nucleoprotein as an antigenic protein or some part of the protein for eliciting the response against the virus.

the result and discussion sections of the manuscript is properly written.

---

## Round 0.2 · accepted · Accept

The authors have properly performed their revision.

Reviewer 1 ·

Basic reporting

All the concerns raised have been addressed by the authors in the revised version of the manuscript.

Experimental design

The revised manuscript has addressed all the concerns raised in this section of the review.

Validity of the findings

The revised manuscript has addressed all the concerns raised in this section of the review.

Additional comments

The authors have adequately addressed all the concerns raised in the review. I appreciate the authors for their time and effort.

·

Basic reporting

no comment.

Experimental design

no comment.

Validity of the findings

no comment.

Additional comments

The authors have addressed my original concerns and substantially revised their manuscript. I think it is much improved.
The only minor issue is that the authors used only 2M NaCl for the dissociation of RNA. This could be further addressed in their future studies.

·

Basic reporting

The Authors have answered all the queries.

Experimental design

No Comment

Validity of the findings

No Comment